# Controlled Release of β-CD-Encapsulated Thyme Essential Oil from Whey Protein Edible Packaging

Andreea Lanciu Dorofte [1], Cristian Dima [1], Alina Ceoromila [2], Andreea Botezatu [3], Rodica Dinica [3], Iulia Bleoanca [1,*] and Daniela Borda [1,*]

[1] Faculty of Food Science and Engineering, Dunarea de Jos University of Galati, 800201 Galati, Romania
[2] Cross-Border Faculty, Dunarea de Jos University of Galati, 111 Domneasca Str., 800201 Galati, Romania
[3] Faculty of Science and Environment, Dunarea de Jos University of Galati, 800201 Galati, Romania
* Correspondence: iulia.bleoanca@ugal.ro (I.B.); daniela.borda@ugal.ro (D.B.); Tel.: +40-336-130-177 (I.B. & D.B.)

**Abstract:** Whey edible films (EFs) functionalized with essential oils have a high potential to be used on various foods due to their antimicrobial and antioxidant activities. Encapsulation is applied for a better retention of volatiles in EFs; however, the functional properties of EFs are modified. The properties of EFs containing thyme essential oil (TEO) encapsulated by co-precipitation in β-CD, developed in three formulae, with inclusion complexes (EF/IC1, EF/IC2, and EF/IC3, respectively) in 15:85, 26:74, and 35:65 mass ratios were studied. Thymol is the main volatile in the ICs with TEO/β-CD (50%–60% of the total volatiles). In comparison with EF/TEO, all three formulae with EF/ICs had better, but similar, WVPs ($p > 0.05$). The EF/IC2 displayed a different FTIR profile than EF/IC1 and EF/IC3, suggesting a smaller number of free functional groups. The EF/IC2 showed better transparency in comparison with EF/IC1 and EF/IC3. All EF/ICs moderately inhibited *R. glutinis*, with the strongest activity registered by EF/IC3 ($p < 0.05$), but did not inhibit *G. candidum*, while showing a strong antibacterial activity against *B. cereus*. All EF/ICs inhibited 65%–70% of the total free radicals. The EF/ICs ensured a gradual release of VOCs in food simulants, with a higher rate in 95% ethanol than in water. These results have demonstrated the properties of EF/ICs with TEO/β-CD as bioactive packaging systems for foods.

**Keywords:** food active packaging; encapsulation by co-precipitation; thyme essential oil; SPME GC/Ms; antimicrobial; antioxidant activity

## 1. Introduction

New food packaging solutions have been studied in recent years as alternatives to conventional packaging, striving to combine biodegradability and sustainability principles backed by consumers with the ability to provide safe food while maintaining its quality during the extended shelf life. To date, promising research results have been published on food edible packaging [1–3]. The emerging applications in edible packaging are related to their versatility in terms of economically efficient use of industrial by-products while reducing waste as well as their ability to be functionalized with a wide range of active substances (e.g., antimicrobials, antioxidants, nutraceuticals) [4,5]. Edible packaging can be used either in the form of thin layers of edible films (EFs) or coatings made of polysaccharides, proteins, lipids, or a combination thereof.

Antimicrobial packaging upscales the traditional impregnation of the food product with herbs and spices by using more concentrated extracts in the form of essential oils (EOs) as efficient substitutes for chemically synthetized preservatives [6]. Essential oils' antimicrobial activity has been extensively studied, succeeding both against Gram-positive bacteria and Gram-negative bacteria, as well as against fungi [7,8]. Active packaging can prevent, for example, foodborne illnesses due to the release of antimicrobials against foodborne pathogens like *Listeria monocytogenes*, *Salmonella enterica*, *Escherichia coli*, and

*Bacillus cereus* [9] or against food spoilage microorganisms like *Pseudomonas*, *Clostridium*, *Geotrichum*, or *Mucor* spp., mostly involved in limiting dairy products' shelf life. Moreover, EOs have been reported as a rich source of antioxidants which could fight cellular oxidative stress effects [10]. For example, Liu et al. [11] reported that konjac glucomannan EF with 1.6% TEO exerted a higher antibacterial effect against Gram-positive bacteria (*L. monocytogenes* and *S. aureus*) than against Gram-negative bacteria (*E. coli O157:H7*), and a 1.5 times increase of the antioxidant effect of the EF with 1.6% TEO compared to the control film, without TEO, due to the high phenolic content present in TEO. Bleoanca et al. [12] tested the antimicrobial activity by vapor phase test of 2.5% (*w/w*) TEO volatiles in whey protein EFs and concluded their effective antimicrobial activity against *Bacillus subtilis*, *Geotrichum candidum*, and *Torulopsis stellata*.

However, the EOs' high volatility, oxidizability, and thermosensitivity call for alternative protection techniques to counteract these negative traits. Several techniques are available for EO protection that are already used for active packaging, like their inclusion in composite materials (e.g., zeolite-thymol antimicrobial composites [13]); nanoemulsion formation (e.g., EF/TEO nanoemulsion used for zucchini active coating [14]); thymol nanoemulsion included in gelatin films [15] or encapsulation (e.g., edible coating with encapsulated TEO in liposomal chitosan applied to cheese [16]); and antimicrobial packaging with coriander EO encapsulated in cyclodextrin nano-sponges [17].

The encapsulation technology employed in EFs could be a solution both for carrying protected bioactive compounds like EOs and for their controlled release to ensure food safety. Several encapsulation techniques have been developed and tested on EOs, like spray-drying [18], complex coacervation [19], electrospinning [20], liposome formation, and solid lipid nanoparticles produced by high-pressure/high-speed homogenization [21]. As one of the most used encapsulant materials, β-cyclodextrin (β-CD) has the capacity of selectively encapsulating lipophilic molecules of EOs inside the truncated cone surface, thus protecting their bioactive properties.

Whey proteins recuperated from whey, a by-product result during cheese processing, have the ability to easily form films. The thermal treatment applied to whey protein solutions triggers disulfide bonds' interchanging and conformational modifications of the three-dimensional structure, enabling the formation of EF packaging. The resulting whey protein EFs provide valuable nutrients, are flexible, can prevent food moisture loss, act as an oxygen barrier, and have neutral taste and flavor; however they exhibit rather poor barrier properties against gases, aromas, and water vapors in comparison with conventional polymers [1]. Whey EFs can be functionalized with EOs and blended with other biopolymers to form composite materials with improved properties. For example, Kontogianni et al. [22] functionalized whey protein concentrate EF with aqueous rosemary and sage extract and successfully applied it to soft cheese stored for 60 days in cold storage at 4 °C. An updated review on active ingredients used on whey protein EFs is presented by the research group of Kandasamy et al. [23]. Another reported successful example refers to the antioxidant nanocomposite packaging obtained by thermoplastic extrusion of corn starch and whey protein, functionalized with rosemary EO [24].

To mitigate EO volatiles' loss, TEO, which has a strong antibacterial, antifungal, and antioxidant reported activity [5,12,25,26], was encapsulated in β-CD prior to its use in whey EFs, taking advantage of both the encapsulant's hydrophobic cavity and hydrophilic properties of the surrounding walls [27].

Therefore, the objective of the current study was to characterize and mutually compare the morphological and physical properties of whey EFs with encapsulated TEO/β-CD powders with three different ICs characterized by GC/Ms. Antimicrobial and antiradical properties of EF/ICs and the release of the volatiles from EFs in two simulants were also evaluated.

## 2. Materials and Methods

Thyme essential oil (TEO) with thymol (*Thymus vulgaris*) obtained by distillation was purchased from Aroma Zone (Provence, France), and β-CD was purchased from Merk KgaA Co., Germany. ProMilk 802FB whey protein concentrate was kindly provided by KUK–Romania (composition on a dry weight basis: protein content 77%, 1% total fat, 11% lactose, 2.9% total ash, 5% moisture). Anhydrous glycerol (purity 99%) and Tween 80 were purchased from Sigma-Aldrich (Bucharest, Romania); 2,2-diphenyl-2-picrylhydrazyl (DPPH) was obtained from Sigma-Aldrich, Germany; 2-octanol was obtained from Sigma-Aldrich Chemie GmbH, Steinheim, Germany; helium (99.996% purity) was obtained from Linde Gaz S.R.L., Galati, Romania; and test microorganisms (*Bacillus cereus* ATCC 10876, *Geotrichum candidum*, and *Rhodotorula glutinis)* were obtained from MIUG collection- Dunărea de Jos University of Galaţi, Romania. The reagents used were of analytical purity and ultrapure water was used to prepare the aqueous solutions.

### 2.1. Encapsulation of Thyme EO (TEO)

Thyme essential oil/β-CD (TEO/β-CD) inclusion complex (IC) was obtained by the co-precipitation method proposed by Petrović et al. [28] and Dima et al. [29], using mixtures of thyme essential oil and β-CD in a mass ratio (TEO/β-CD) of 15:85, 26:74, and 35:65. We had previously optimized the TEO/β-CD mass ratio within the EF/ICs powders and estimated the efficacy and yield of the EFs (data not shown).

Briefly, an amount of 8.5 g of β-CD was solubilized in an alcoholic solution (70%) and heated to a temperature of 55 °C. Later, different amounts of TEO were added—1.5, 3, and 4.5 mL—to obtain powders with IC1, IC2, and IC3, respectively. The solution was stirred for 4 h at room temperature and then stored in the refrigerator for later filtration. The filtration was carried out under vacuum and the TEO/β-CD ICs were dried for 24 h at 55 ± 0.5 °C and RH = 65%, thus obtaining fine white powder. The powder was stored in dark plastic containers at 22 ± 0.5 °C and analyzed from physicochemical and morphological points of view.

### 2.2. Edible Films with Encapsulated CD-TEO (EF/CD-TEO)

First, 8% of whey protein concentrate (WPC) was dispersed in ultrapure water at room temperature under continuous magnetic stirring (250 rpm, 25 min). In order to form flexible films, a thermal denaturation was performed at 80 ± 0.5 °C, for 35 min. The denaturation was stopped by cooling the sample in iced water, following the method previously optimized by our research group [14]. Anhydrous glycerol (7.9% *w/w*) was added as a plasticizer and 1.6% (*w/w*) Tween 80 as a stabilizer.

Film-forming solutions were functionalized with TEO due to its antimicrobial activity related to the high content of effective antimicrobial terpenes [30]. To obtain a stable and homogeneous emulsion before EF casting, sonication (35% amplitude, 3 min) was performed with Sonopuls HD3100 equipment (Bandelin, Germany) using a sonication probe of 8 mm diameter. The temperature of the film-forming solution was kept constant during sonication, at 23 ± 2 °C, by placing the sample in an iced water bath. The film-forming emulsions were further subjected to degassing for 3 min to eliminate air bubbles.

Films with TEO were cast in two variants: films with non-encapsulated 2.5% *w/w* TEO (EF/TEO), and films with TEO encapsulated in beta-cyclodextrin (EF/IC) in three formulae with 2.5% inclusion complexes (ICs): EF/IC1 (15% TEO in β-CD), EF/IC2 (26% TEO in β-CD), and EF/IC3 (35% TEO in β-CD). A film containing β-CD (EF/β-CD) was used as the control. For IC, the TEO/β-CD powders were added to the protein solution after sonication to protect the oil capsules from disintegration.

The films were dried at RT for 48 h and further preconditioned at 25 ± 1 °C, 50% RH using a solution of propylene glycol placed in a glass desiccator for 24 h. All experiments were performed in triplicate.

### 2.3. Characterization of ICs

*Volatile Fingerprint of ICs*

The GC profile of the volatiles' presence in the EFs was evaluated using 2-octanol as an internal standard. A Trace GC–MS Ultra with ion trap MS ITQ 900 (Thermo Scientific, Waltham, MA, USA) equipped with a solid phase microextraction (SPME) system having a DVB/CAR/PDMS fiber was used for analysis. The method of incubation, extraction, and analysis was previously optimized and described in detail [30]. The column used for separation was the TG-WAX capillary column (60 m × 0.25 mm, i.d. 0.25 μm). The mobile phase was helium (99.996% purity, Linde Gaz S.R.L., Galati, Romania) maintained at a constant flow rate of 1 mL/min during analysis.

Samples of 0.1523 ± 0.0010 g of IC powders were weighed in glass vials (20 mL), and further spiked with 10 μL 2-octanol 0.651 mg/mL (Sigma-Aldrich Chemie GmbH, Steinheim, Germany) added as internal standard and then incubated at 50 °C for 20 min. After extraction in the gaseous phase and 4 min splitless desorption into the GC injection port, the compounds were exposed to the temperature ramp and further separated on the column. The ramp selected was 50 °C/4 min followed by an increase to 75 °C with 3 °C/min, then at 120 °C with 5 °C/min, to 170 °C at 4 °C/min, and to 220 °C at 10 °C/min, when the temperature was kept constant for 10 min. The temperature of the transfer line in MS was set to 250 °C. Mass spectra were obtained from the positive ions resulting from scanning in the 50 to 650 m/z range and 200 eV.

The volatile organic compounds (VOCs) were tentatively identified using the NIST 08 library database available within Xcalibur software and reported as a percentage of the individual peak area from the total VOCs.

### 2.4. Morphological Properties of EFs

#### 2.4.1. Structure of EFs by SEM

The morphological analysis of the composite films was performed by scanning electron microscopy SEM (Quanta 200, former FEI, now Thermo Fisher Scientific, Waltham, MA, USA) at an accelerating potential of 15 kV. As a pre-processing step, the samples were glued onto an aluminium strip and then vacuum coated with a thin layer of gold. Then, SEM images were taken, and analyzed at 5000× magnification.

#### 2.4.2. Fourier Transform Infrared Spectroscopy (FT-IR) Analysis

The EFs were directly analyzed with a Nicolet iS50 FT-IR spectrometer (Thermo Scientific, Waltham, MA, USA) equipped with attenuated total reflection (ATR) accessory, DTGS detector, and KBr beamsplitter. The FT-IR spectra were collected after co-adding and averaging 32 scans over the 4000–400 cm$^{-1}$ range with 4 cm$^{-1}$ resolution. A background spectrum of air was considered before each sample processing. The ATR plate was cleaned with ethanol after each spectrum, and a background spectrum was collected and compared to the previous background spectrum to ensure that no residue was present. The environmental temperature was kept constant (21 °C) during the measurements.

### 2.5. Physical Properties of EFs

#### 2.5.1. EF Thickness

The thickness of the films was determined with an electronic caliper (Burg Wachter, Germany) with a measurement range of 0–150 mm and a precision of 0.02 mm. Ten measurements were taken for each sample in randomly chosen areas of each film and the average value ± stdev was reported as film thickness.

#### 2.5.2. EF Dry Matter

The dry matter (DM) content was determined using the gravimetric method by differential weighing of the film sample before and after drying. Briefly, 1 g of each film sample

was dried in an oven at $105 \pm 1\,°C$ until constant weight was attained. Dry matter content (%) was calculated using Equation (1):

$$DM = \frac{m_2 - m_0}{m_1 - m_0} \times 100, \ \ \%$$

(1)

where $m_0$—weight of empty and dried glass bottle, (g); $m_1$—weight of the glass bottle with film, before drying, (g); and $m_2$—weight of the glass bottle with film, after drying, (g).

### 2.5.3. Water Activity

The water activity (aw) of preconditioned edible films was measured with a Fast lab water activity meter (GBX, Loire, France), using discs of films ($4 \pm 0.1$ cm diameter).

### 2.5.4. EF Transparency

The transmittance and the opacity of the edible films were determined using a UV–visible spectrophotometer (Cintra 202, GBC Scientific Equipment Ltd., Braeside, Australia). The film samples were cut into rectangular pieces (1.0 cm $\times$ 4.5 cm) and placed on the sample compartment of a spectrophotometer. An empty compartment was used as a blank reference in the measurement. The opacity of the films was calculated according to the method described by Bai et al. [30] using Equation (2):

$$Opacity = \frac{Abs600}{X}$$

(2)

where Abs600—absorbance at 600 nm and X—film thickness (mm).

### 2.5.5. Water Vapor Permeability of EFs

Water vapor permeability (WVP) was determined by gravimetric analysis. Films' discs of $49.82 \pm 0.29$ mm diameter were equilibrated at $25 \pm 0.5\,°C$, 50% $w/w$ RH using a solution of propylene glycol placed in a glass desiccator, for 48 h. The discs were sealed onto 50 mL polypropylene vials filled with distilled water (100% RH) to 10 mm below the inner face of the film. The vials had an inner diameter of 29 mm and a height of 115 mm. The vials with attached films were kept at a temperature of $25 \pm 0.5\,°C$ in a glass desiccator with 33% RH humidity provided by the $MgCl_2$ saturated salt solution.

The permeability of the films was calculated according to the method described by Bleoanca et al. [12] using Equation (3). The weight loss of the films was calculated as the slope of the linear regression equations where the coefficient of determination ($R^2$) was >0.99.

$$WVP = \frac{Slope \ \times \ x}{A \ \times \ \Delta p}, \ g \cdot mm/m^2 \cdot s \cdot Pa$$

(3)

where slope—weight loss of the falcon with water per second, (g/s); x—average thickness of the film, (mm); A—area of the exposed film, ($m^2$); and $\Delta p$—vapor pressure difference across the test film, (Pa).

### 2.5.6. Swelling Capacity of EF

The swelling index (SI) was determined using the modified method described by Galus et al. [31]. The equilibrated films were cut into discs of 10 mm diameter, weighed ($m_0$), and then immersed in distilled water at $25\,°C$ for 2 min. After removal from the water, the films were blotted with filter paper to remove excess water and weighed again ($m_1$). The amount of water absorbed was calculated by subtracting the initial dry films' weight of from the weight of the wet samples ($m_1 - m_0$) and expressed as a percentage of the initial weight of the films (Equation (4)):

$$SI = \frac{(m_1 - m_0)}{m_0}, \ \%$$

(4)

*2.6. Bioactive Properties*

2.6.1. Antimicrobial Activity of EFs

EFs containing encapsulated TEO (IC1, IC2, and IC3) were analyzed for antimicrobial activity by disk diffusion assay [28] against three test microorganisms, *Bacillus cereus* ATCC 10876, *Geotrichum candidum*, and *Rhodotorula glutinis* from the MIUG collection (Dunarea de Jos University of Galati, Romania). Plate count agar was used for antibacterial assay and rose-bengal chloramphenicol agar (Oxoid Ltd., Hampshire, UK) was used for anti-yeast and fungi growth evaluation. The inoculum of each test microorganism was prepared by overnight culturing of single colonies at either 37 °C for bacteria or 27 °C for yeast and fungi. Films of 5 cm diameter were placed onto appropriate culture media for the test microorganisms inoculated with approximately $10^6$ CFU/mL of the three test microorganisms. Following appropriate incubation, inhibition diameters that developed around the film disks were measured with a digital caliper (Burg Wachter, Wetter, Germany). The results of three independent experiments are reported as average values of the extent of inhibition zones $\pm$ standard deviation.

2.6.2. DPPH Radical Scavenging Activity

The antioxidant properties of EF/IC were evaluated by the 2,2-diphenyl-1-picrylhydrazyl (DPPH) radical scavenging assay. The extraction of the TEO from films' samples (0.1 g) was made in 2 mL of methanol by vortexing (2 min). Then, the mixture was further centrifuged for 30 min at 8000 rpm and the supernatant was collected to be used for the radical scavenging assay. Test samples were prepared by mixing 0.5 mL of the supernatant with 2 mL of methanolic solution of DPPH (0.063 mM) and 1 mL of methanol [32]. After shaking with a vortex, the mixture was kept in the dark at room temperature for 30 min and, afterwards, the absorbance of the solution at $\lambda_{max}$ = 517 nm was recorded using a UV-VIS spectrophotometer (UV-VIS Cintra 202, GBC Scientific Equipment Ltd., Braeside, Australia). The results were expressed as % of radical scavenged activity using Equation (5):

$$\text{Antiradical activity } (\%) = \frac{A_C - A_S}{A_S} \times 100 \qquad (5)$$

where $A_C$ and $A_S$ are the absorbance of the free DPPH solution and the absorbance of the DPPH solution with EF/IC, respectively.

*2.7. TEO Release Test in Food Simulants*

The release rate of TEO from EF/ICs was determined by using two different food simulants—water and ethanol 95% and a lipophilic foods simulant—both selected based on their different polarity [33].

The TEO release rate was determined based on the adapted method presented in two studies [34,35]. Briefly, $2 \times 2$ cm$^2$ EF/ICs immersed into 10 mL of each food-simulating solvent were continuously agitated at 25 °C, 150 rpm for different contact times depending on the solvent simulant. At pre-determined time intervals (0, 3, 6, 10, 24, 28, 72 h), 3 mL of the supernatants were used for spectrophotometric analysis at 370 nm using the spectrophotometer UV-VIS (Cintra 202, GBC Scientific Equipment Ltd., Braeside, Australia) and returned to the vials. Concentration of released TEO was estimated using a standard curve. All measurements were performed in three replicates per time and per sample.

*2.8. Statistical Analysis*

All data are the means of three replicates $\pm$ standard deviation (SD). Analysis of variance (ANOVA) and Tukey's post hoc test were applied to evaluate significant differences among groups ($p < 0.05$).

The spectra resulting from the FT-IR analysis were pre-processed using multiplicative scatter corrections (MSC) and overlapped. The contribution of the main factors in explaining the total variance was evaluated by Principal Component Analysis (PCA) with Varimax

rotation method, considering the reduction of total variance as a cut-off criterion in PCA. The Unscrambler software (Version 9.7; CAMO, Oslo, Norway) was used for PCA.

## 3. Results

### 3.1. Volatile Fingerprint of TEO/β-CD Inclusion Complexes

A "host-guest" relationship, specific to a stable structure called the *inclusion complex* [22], results from the interactions between the β-CD molecules and the lipophilic ones of TEO.

In the present study, the thyme essential oil was encapsulated in β-CD by means of the co-precipitation method and further included in EFs. For this, different β-CD: TEO mass ratios were used—15:85, 26:74, and 35:65 in IC1, IC2, and IC3 powders, respectively. Since thymol is the main component in the thyme essential oil, it is considered that the cavities of the β-CD molecules are preferentially occupied by the thymol molecules. The presence of the hydroxyl group in the structure of the thymol molecule and other terpenes also generates a potential accumulation of volatiles on the outer surface of the truncated cone forming hydrogen bonds distributed on the outer surface of the β-CD molecule. The presence of volatiles on the outer surface of the β-CD molecule with weaker bonds makes VOCs more prone to volatilization than the one inside the β-CD.

The volatile profile of *β-CD* powders with ICs was characterized using SPME GC/MS with a total of 21 VOCs tentatively identified using the NIST library and the resulting ions, indicated in Table 1.

**Table 1.** The GC/MS SPME fingerprint of TEO ICs and the % of volatiles from the total VOCs in the analyzed powders.

| Compound | KI | Ions | IC1 | IC2 | IC3 |
|---|---|---|---|---|---|
| Thujene | 825 | 91; 77; 93; 51; 136 | 0.15 ± 0.05 d,B * | 0.46 ± 0.02 e,A | 0.54 ± 0.02 e,A |
| α Pinene | 925 | 91; 93; 136; 121; 93 | 1.51 ± 0.10 c,d,A | 0.74 ± 0.12 e,B | 0.80 ± 0.24 e,B |
| Ciclohexene 1-methyl, 4-methyldiene | 939 | 91; 77; 93; 51; 107 | 0.37 ± 0.05 d,C | 1.26 ± 0.07 e,B | 1.51 ± 0.09 e,A |
| 1-S alpha Pinene | 953 | 91; 67; 93; 94; 65 | 0.08 ± 0.01 d,C | 0.21 ± 0.01 e,B | 0.24 ± 0.01 e,A |
| 3-Carene | 993 | 91; 93; 136; 77; 121 | 3.33 ± 0.15 c,d,C | 10.18 ± 0.87 c,B | 12.07 ± 0.95 c,A |
| Ocymene | 1013 | 119; 91; 134; 135; 169 | 7.15 ± 0.52 b,C | 19.57 ± 1.11 b,B | 23.08 ± 1.54 b,A |
| Isoledene | 1114 | 105; 119; 169; 91; 79 | 0.01 ± 0.00 d,A | 0.02 ± 0.00 e,A | 0.02 ± 0.00 e,A |
| Carenol | 1122 | 67; 79; 81; 91; 109 | 1.04 ± 0.05 c,d, A | 0.04 ± 0.00 e,B | 0.04 ± 0.00 e,B |
| Copaene | 1123 | 105; 119; 161; 91; 79 | 0.03 ± 0.00 d,A | 0.03 ± 0.00 e,A | 0.02 ± 0.00 e,A |
| a-Panasinsene | 1137 | 91; 95; 67; 105; 119 | 1.01 ± 0.08 c,d,B | 1.41 ± 0.10 e,A | 1.32 ± 0.09 e,A |
| Camphene | 1144 | 91; 93; 77; 121; 67 | 0.36 ± 0.05 d,B | 0.59 ± 0.02 e,A | 0.53 ± 0.02 e,A |
| 4-Carene | 1169 | 91; 105; 133; 77; 161 | 0.73 ± 0.04 d,A | 0.62 ± 0.03 e,B | 0.63 ± 0.03 e,B |
| a-Phellandrene | 1170 | 91; 93; 77; 111; 67 | 0.04 ± 0.01 d,A | 0.04 ± 0.01 e,A | 0.03 ± 0.01 e,A |
| Longifolene | 1186 | 91; 105; 119; 161; 77 | 0.00 ± 0.00 d,A | 0.01 ± 0.00 e,A | 0.00 ± 0.00 e,A |
| trans-Chrysanthemal | 1187 | 81; 123; 95; 67; 166 | 0.01 ± 0.00 d,A | 0.00 ± 0.00 e,A | 0.01 ± 0.00 e,A |
| Seychellene | 1295 | 91; 93; 77; 105; 121 | 0.04 ± 0.00 d,A | 0.00 ± 0.00 e,A | 0.00 ± 0.00 e,A |
| 2-Bornene | 1223 | 91; 93; 121; 79; 136 | 0.06 ± 0.02 d,A | 0.14 ± 0.06 e,A | 0.11 ± 0.02 e,A |
| Borneol | 1306 | 95; 67; 121; 65; 136 | 0.03 ± 0.00 d,B | 0.06 ± 0.01 e,A | 0.04 ± 0.01 e,A,B |
| Epizonarene | 1314 | 119; 161; 105; 204; 91 | 0.22 ± 0.03 d,A | 0.07 ± 0.02 e,B | 0.07 ± 0.02 e,B |
| Thymol | 2097 | 136; 150; 105; 107; 91 | 65.90 ± 5.35 a,A | 54.36 ± 4.21 a,B | 49.45 ± 3.96 a,B |
| Carvacrol | 2381 | 135; 150; 108; 92; 79 | 4.5 ± 0.21 b,c,B | 6.12 ± 0.52 d,A | 5.57 ± 0.41 d,A |

* different small caps letters indicate significant differences ($p < 0.05$) on columns and capital letter on rows, estimated by unifactorial ANOVA and post hoc Tuckey test; KI-Kovats Index; Data re-ported are mean ± SD of three replicates.

The main volatile present in all powders was thymol, identified in % varying from ~50% in IC3 to ~66% in IC1. The high content of thymol in IC1 could be associated with a higher encapsulation rate in IC1 compared to IC2 and IC3. Other major compounds were o-cymene, carvacrol 3-carene, and α-pinene. The different % of volatiles in each IC could suggest different properties of the powders depending on the concentration and bioactivity of the active molecules. Thymol was the major constituent of *T. vulgaris* (53%) found in

β-CD powders by Toro-Sanchez et al. [36]; however, other prevalent compounds such as p-cymene, γ-terpinene, and linalool were not identified in the current study.

The highest percentage of thymol was present in β-CD/IC1, and it was significantly higher ($p < 0.05$) in comparison with β-CD/IC2 and β-CD/IC3. This could be explained by the reduced inclusion efficiency of IC2 and IC3 but also by a competition in occupying the β-CD inner cavity between thymol and other molecules (data not shown). In the case of 3-carene, the % from the total VOCs was significantly ($p < 0.05$) higher in IC2 and IC3 in comparison with IC1. In the case of carvacrol, the %s were very close; however, significant differences ($p < 0.05$) were identified between IC1 and IC2 and IC3.

### 3.2. Morphological Properties

### 3.2.1. SEM Analysis

Microstructure evaluation by scanning the cross-sections of EF/TEO (Figure 1a) indicates the presence of TEO droplets as lighter spheres into the darker EF matrix, with an average diameter of $1.38 \pm 0.24$ μm. During the electron beam interaction, and due to their different chemical natures, the TEO, ICs, and EF appear in various gray tones and morphologies in the SEM images. Therefore, three types of structure can be observed: the TEO components as whiter spheres (Figure 1a) and the IC inclusions like rods (Figure 1c)—both having the crystalline structure—which are incorporated/encapsulated into the EF matrix, as a continuous, homogeneous film (amorphous structure).

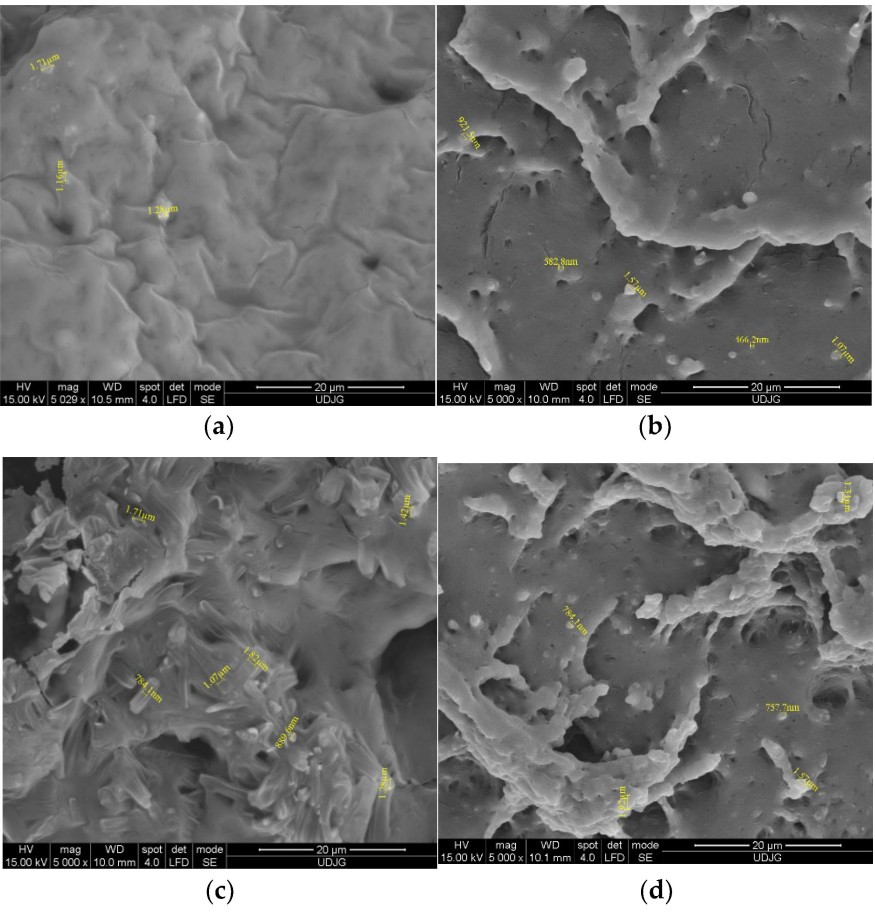

**Figure 1.** SEM analysis of films EF with TEO (**a**) and with IC1 (**b**); IC2 (**c**) and IC 3 (**d**).

The SEM evaluation of the EF/ICs' cross-section (Figure 1b–d) revealed the presence of typical monocyclic crystalline structures, characteristic to β-CD encapsulant, and similar to other EO/β-CD complexes reported by Ghadetaj et al. [32] for example. The inclusion complexes have diameters ranging from 466.2 nm (IC1) to 1.92 μm (IC3), the majority

with a nanodiameter. Figure 1c best presents the cross-section of EF/IC2, with numerous rectangular encapsulant nanostructures and TEO droplets.

### 3.2.2. FT-IR Analysis

As can be seen from Figure 2, all the EF/ICs have fewer peaks in comparison with β-CD due to the loss of vibration in the free groups present on the outer surface of β-CD and different complexes' formation [36].

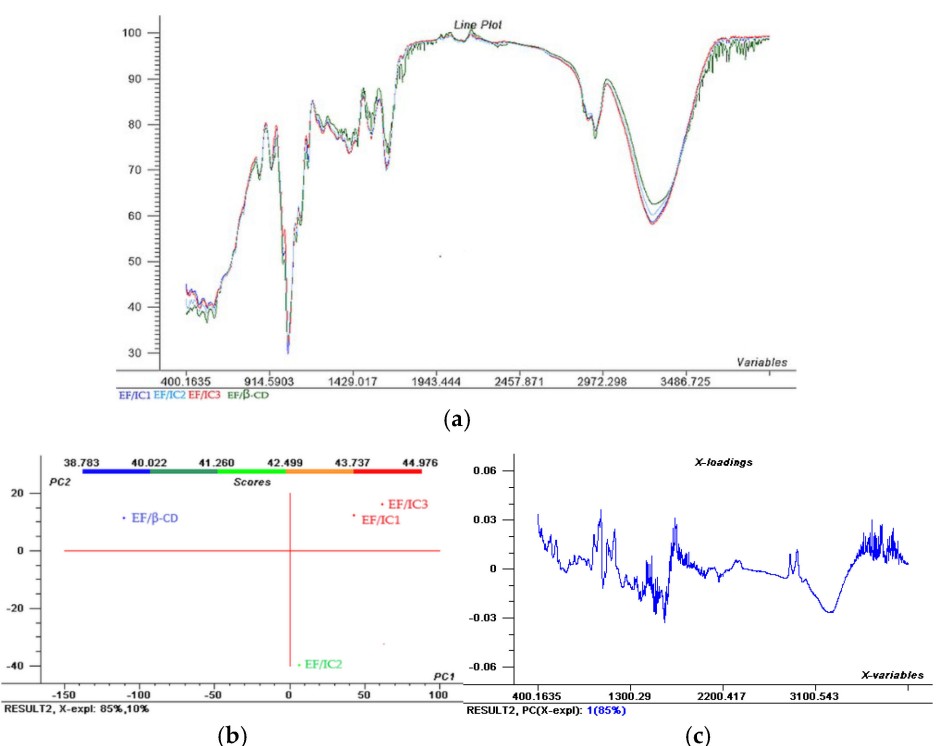

**Figure 2.** The Aligned FTIR Spectra (**a**) and Scores; (**b**) representation of the two principal components (PC1 and PC2) for EF/IC1, EF/IC2, EF/CI3 and control EF/β-CD and & Loadings; (**c**) spectra for EF/IC1.

Figure 2 reveals that, at 3293 cm$^{-1}$, broad peaks corresponding to O–H stretching are present in the EF with IC1, IC2, and IC3 and with β-CD, specific for alcohol's presence. The highest intensity of O-H bonds were present in EF/IC3 and EF/IC1, suggesting a higher number of O-H bonds in comparison with EF/IC2, while the fewest O-H bonds were found in the films loaded with β-CD, considered the control.

At 2970 cm$^{-1}$, absorption bands are specific for C-H stretching vibration and the bands are overlapped for all EF/ICs with the highest intensity for EF/β-CD [37].

According to Rezaei et al. [38], the peaks between 1450 and 1600 cm$^{-1}$ (at 1639, 1539, 1506, and 1410 cm$^{-1}$) corresponded to the characteristic peaks of thymol and may be related to the stretching of the –CH$_2$– group.

The Scores in Figure 2b show that most of the variation is explained by PC1 (85%) and 10% is explained by PC2. In the 1st quadrant, EF/IC1 and EF/CI3 are present and showing many similarities of the whole FTIR spectra, while in the 2nd quadrant, the control has different characteristics. Interestingly, EF/IC2 shows a dissimilar spectrum compared to EF/IC1 and EF/IC 3, being positioned in a different cluster, in the 4th quadrant that could suggest fewer free functional groups due to better interaction with the EF biopolymeric composition.

### 3.3. Physical Properties

3.3.1. Film Thickness

The EF with IC obtained had thicknesses ranging between 0.362 and 0.370 mm (Table 2) with no significant differences ($p > 0.05$). A wide range of biopolymer film thicknesses has been reported in the literature, indicating its dependence on both film composition and processing parameters [39] and, in this case, on the volume per surface considered when casting the EFs.

**Table 2.** Dry matter, thickness, and water activity of EFs.

| Film | Thickness (mm) | DM (%) | $a_w$ |
|---|---|---|---|
| EF/IC1 | 0.362 ± 0.034 [b] * | 84.467 ± 8.393 [a] | 0.533 ± 0.052 [c] |
| EF/IC2 | 0.370 ± 0.038 [b] | 83.661 ± 8.402 [a] | 0.552 ± 0.051 [c] |
| EF/IC3 | 0.366 ± 0.035 [b] | 83.565 ± 8.326 [a] | 0.539 ± 0.054 [c] |
| EF/TEO | 0.314 ± 0.033 [b] | 82.296 ± 8.224 [a] | 0.507 ± 0.054 [c] |

* The values obtained are the mean ± SD of three analyses. Different letters obtained by ANOVA and Tukey's post hoc test at $p < 0.05$ indicate significant differences between samples.

3.3.2. Dry Matter

Dry matter is an important characteristic of films. The dry substance plays an important role in evaluating the performance of the film, since a low value of the dry substance and, implicitly, a high value of the film's humidity can change the properties of the film and of the food covered by film, due to migration or moisture transferring [40]. As seen in Table 2, the dry matter content of the EF film with TEO was 82.296 ± 8.224 while the DM slightly increases with the addition of ICs or β-CD to the control film. However, no significant differences were registered between the DM of edible films with ICs or with β-CD ($p > 0.05$).

3.3.3. Water Activity

The results obtained are presented in Table 2. The addition of unencapsulated TEO determined the lowest water activity of all analyzed films, with no significant differences recorded ($p > 0.05$).

3.3.4. Transparency

The transparency of a film is defined as "its ability to transmit light" with the mention that this attribute "correlates with its usual transmittance" definition given by the American Society for Testing Materials [41]. Thus, the transparency of a film is measured based on the transmittance of light through the film, while opacity is often measured based on the light absorbed by the material being analyzed.

The light-visible (VIS) barrier properties of the film were measured at various wavelengths (in the range 450–650 nm), using a double-beam UV-VIS spectrophotometer. The values 650 nm, 550 nm, and 450 nm refer to the peak wavelengths of visible light to which the color sensors respond. These wavelengths correspond approximately to red, green, and blue.

In the 450–650 nm VIS-light wavelength range, the transmittance varied from 23.08 ± 2.25 to 28.29 ± 2.82% in EF/TEO; from 20.31 ± 2.01 to 27.47 ± 2.64% in EF/IC1; from 37.11 ± 3.65 to 48.46 ± 4.71% in EF/IC2; and from 29.51 ± 2.84 to 38.27 ± 3.61 in EF/IC3 (Table 3).

The transmittance value increases in all cases with the increase in wavelength in the 450–650 nm range, the strong absorption of light at 650 nm being consistent with literature reports [42]. Transmittance increased 1.23 times with the increase in wavelength from 400 to 650 nm in EF/TEO, 1.35 times in EF/IC1, 1.30 times in EF/IC2, and 1.29 times in EF/IC3. IC films absorb more light than TEO films. Possible explanations for the lower increase in the case of EF/TEO may be the high content of aromatic amino rings in the protein-based structure that can absorb UV radiation [43] and the reflection caused by the free oil droplets' presence. The lowest transparency was displayed, as expected, by the

films that contained only EF/β-CD. In the case of the IC film, the transmittance values are relatively close; however, the EF/IC2 film had the highest transmittance values at all measured wavelengths, indicating good transparency. Since transmittance values were below 90% in all formulations, all films can be considered semi-transparent [43].

**Table 3.** Transmittance (%) of films.

| EFs | λ (nm) 450 | 500 | 550 | 600 | 650 |
|---|---|---|---|---|---|
| EF/TEO | 23.08 ± 2.25 [c] * | 24.79 ± 2.42 [c] | 26.03 ± 2.55 [c] | 27.21 ± 2.62 [c] | 28.29 ± 2.82 [c] |
| EF/IC1 | 20.31 ± 2.01 [c] | 22.53 ± 2.15 [c] | 24.43 ± 2.32 [c] | 26.11 ± 2.45 [c] | 27.47 ± 2.64 [c] |
| EF/IC2 | 37.11 ± 3.65 [a] | 40.85 ± 4.01 [a] | 44.04 ± 4.37 [a] | 46.42 ± 4.55 [a] | 48.46 ± 4.71 [a] |
| EF/IC3 | 29.51± 2.84 [b] | 32.34± 3.01 [b] | 34.81± 3.32 [b] | 36.72 ±3,51 [b] | 38.27 ± 3.61 [b] |
| EF/β-CD | 15.56 ± 1.52 [d] | 17.11 ± 1.68 [d] | 18.22 ± 1.93 [d] | 19.32 ±2.1 [d] | 20.30 ± 1.95 [d] |

* Values obtained are the mean ± SD of three analyses. Different letters obtained by ANOVA and Tukey's post hoc test at $p < 0.05$ indicate significant differences between samples.

The opacity of the EF/IC2 film has the lowest value compared to the other analyzed films. The higher opacity of EF/IC1 may be due to a greater number of β-cyclodextrin molecules loaded with TEO molecules.

Although the powder concentration in the films increased, the opacity did not increase, but decreased in the EF/IC2 and EF/IC3 films in comparison with EF/IC1, probably because of the excess of unencapsulated TEO remaining on the surface of β-CD powder that reflected light and decreased the opacity of the films. However, no significant differences ($p > 0.05$) were observed between EF/IC2 and EF/IC3 (Figure 3). The different opacity values can also be attributed to the agglomerations of the added components in the film matrix, which have the effect of reducing the passage of light through its surface and favoring light dispersion [44].

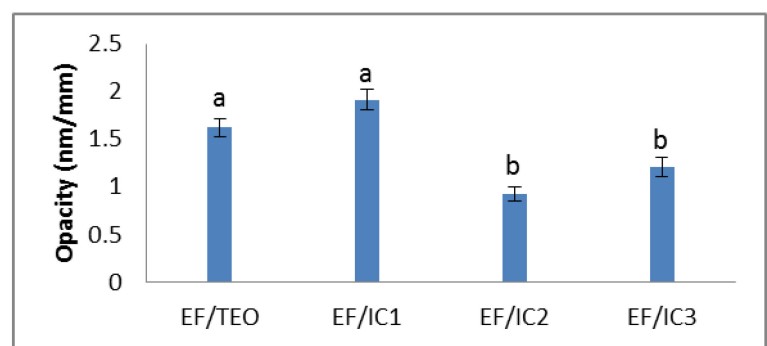

**Figure 3.** The opacity of EF/TEO, EF/IC 1, EF/IC 2 and EF/IC 3 films (nm/mm). Values obtained are the mean ± SD of three analyses. Different letters obtained by ANOVA and Tukey's post hoc test at $p < 0.05$ indicate significant differences between samples.

3.3.5. Water Vapor Permeability (WVP) of EFs

Relative humidity (RH) is an important parameter in WVP calculation and, in this case, a 100%–50% RH (on each side of the film-covered container) gradient is applied on the sides of the film surface. Weight loss of the distilled water containers covered with whey protein films resulted in WVP values calculated from the slopes of the linear regression equations. Part of the water vapor lost will remain absorbed onto the film surface while another part is released into the environment in the form of water vapor (Figure 4).

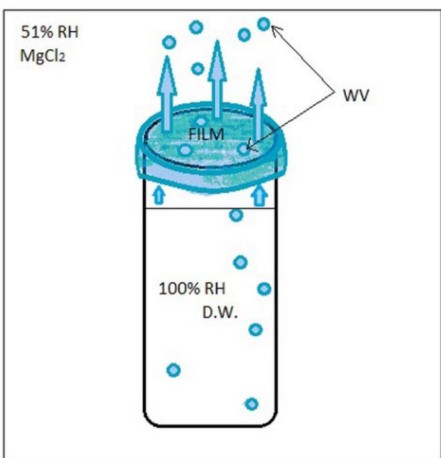

**Figure 4.** Water vapor permeability of edible films.

The WVP results show that the form in which the oil is added to the film matrix, encapsulated or non-encapsulated, influences the water vapor permeability of the film (Figure 5).

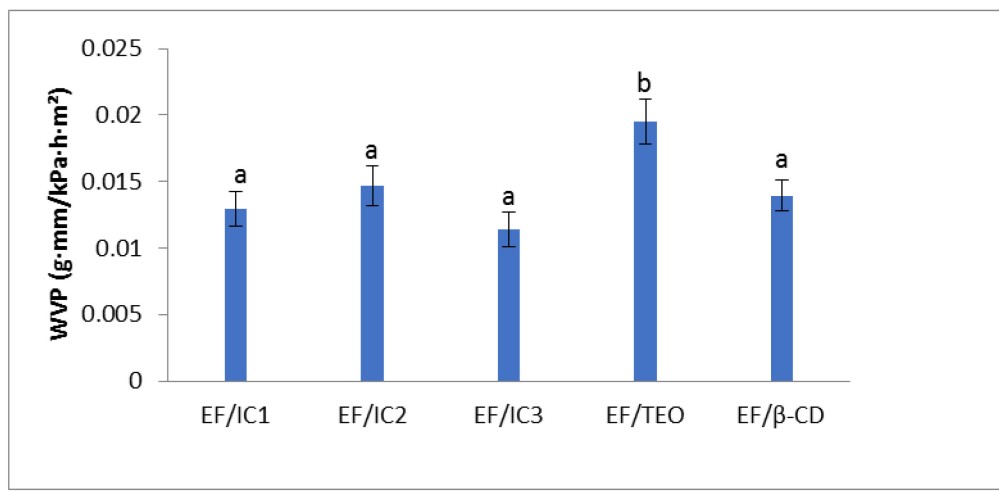

**Figure 5.** Effect of EO added to whey protein films in different forms (inclusion complexes IC1, IC2, IC3, and EF/TEO) versus EF/β-CD film on water vapor permeability. The values obtained are the mean ± SD of three analyses. Different letters obtained with ANOVA method and Tukey's post hoc test at $p < 0.05$ indicate significant differences between samples.

It can be noticed that, compared to EF/TEO, EF/IC1 has a lower permeability to water vapor by 1.506 times, EF/IC2 by 1.329 times, EF/IC3 by 1.707 times, and EF/β-CD by 1.39 times. This surprising behavior was also observed in other studies and its occurrence was explained by the increased density in the constituent elements or their concentration in the film that resulted in changes of the WVP [45]. Thus, formation of a dense network in films due to the introduction of β-CD powders could have reduced the volume occupied by water molecules [46]. No significant differences ($p > 0.05$) between the films containing powders (EF/ICs and EF/β-CD) were observed, while the EF/TEO film is significantly different ($p < 0.05$) from the other analyzed films. The experimental results of WVP obtained by Xu. et al. [46], regarding films with chitosan and cinnamon essential oil stabilized by ethyl-Nα-lauroyl-L-arginate hydrochloride (LAE) alone or co-stabilized by LAE and hydroxypropyl-β-CD were in the same range with our results, between 0.2–0.5 g·mm/kPa·h·m².

### 3.3.6. Swelling Index of EF

The SI values show that the analyzed films have a similar behavior (Figure 6). Similar results were obtained by Galus et al. [30] who analyzed oil-free films compared to films with almond and walnut oil. The value of the swelling index of EF/IC2 and EF/TEO is higher than EF/IC3 and EF/β-CD. The EF/β-CD has a swelling index 1.52 times lower than EF/TEO, which means that the addition of β-CD to the film decreases the amount of water absorbed into the film, similar to EF/IC3. In comparison, Cao et al. [47] found in their study that the protein films analyzed with TEO/β-CD showed lower swelling index values compared to gelatin protein films or protein-gelatin composite films, probably due to differences in the film compositions and the type of aggregates formed within the film structure.

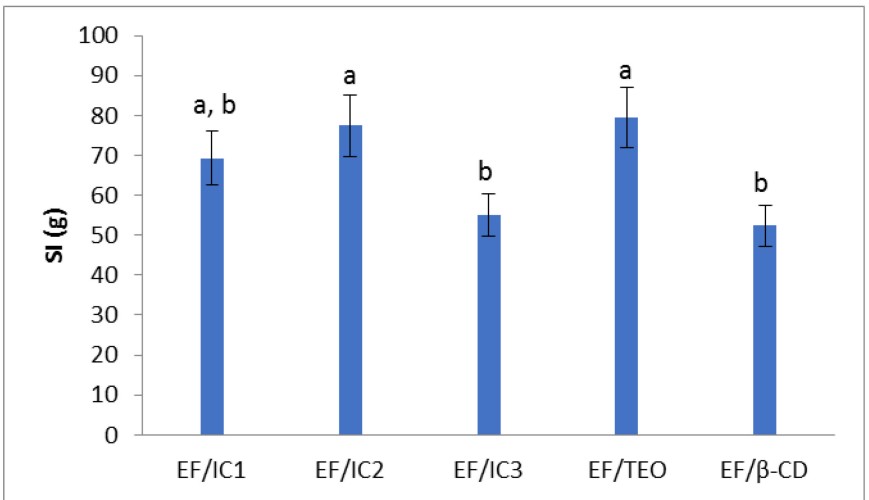

**Figure 6.** The influence of EO added in protein films in different forms (inclusion complexes IC1, IC2, IC3, EF/TEO) compared to oil-free β-CD film on the swelling index. The values obtained are the mean ± SD of three analyses. Different letters obtained by the ANOVA method and Tukey's post hoc test at $p < 0.05$ indicate significant differences between samples.

### 3.4. Bioactive Properties

3.4.1. Antimicrobial Activity of EF

All EFs with unencapsulated TEO had an antimicrobial effect against all test microorganisms, with the inhibition diameters ranging from 20.71 mm (*R. glutinis*) to 47.57 mm (*G. candidum*), significantly different from one to another (Table 4).

**Table 4.** Inhibition zones of EF with unencapsulated TEO (control) and encapsulated TEO (IC1, IC2, IC3). Results are average ± SD of three independent tests, expressed in mm.

| Test Microorganisms | Sample Control EF/TEO | EF/IC1 | EF/IC2 | EF/IC3 |
|---|---|---|---|---|
| *Bacillus cereus* | 38.99 ± 0.36 [a,A] * | 34.09 ± 0.51 [a,B] | 33.47 ± 0.24 [a,B] | 33.98 ± 0.40 [a,B] |
| *Geotrichum candidum* | 47.57 ± 0.25 [b] | ND ** | ND | ND |
| *Rhodotorula glutinis* | 20.71 ± 0.41 [c,A] | 15.63 ± 0.64 [b,B] | 14.20 ± 0.62 [b,B] | 16.44 ± 0.40 [b,C] |

* Different small caps letters indicate significant differences ($p < 0.05$) on columns and capital letter on rows, estimated by unifactorial ANOVA and post hoc Tuckey test; ** ND-not detected.

The antimicrobial activity of encapsulated materials is usually higher than that of free form [48], but there are also exceptions [49], which is also this research case, reporting vice

versa results. The reduced antimicrobial activity exerted by EF/ICs could be explained by the longer time needed for releasing TEO from ICs particles to exert antimicrobial activity [49].

The EF/CD-TEO's antimicrobial effect was classified as strong if the inhibition area ≥20 mm, moderately for 12 mm < inhibition zone < 20 mm, and not inhibitory for inhibition zones < 12 mm [50]. All films with encapsulated TEO exerted a strong inhibition against *Bacillus cereus*, with no significant statistical difference between IC1, IC2, and IC3. *Rhodotorula glutinis* was moderately inhibited by the three films with encapsulated TEO, with no significant difference between EF/IC1 and EF/IC2, while EF/IC3 exhibited the highest inhibition zone, significantly different from the other two EF/ICs. Even though unencapsulated EF/TEO determined the highest inhibition area for *Geotrichum candidum*, encapsulated TEO included in EFs had the opposite effect, with no detectable inhibition area for any of the EF/ICs against the tested mould. Previous antifungal tests (data not shown) performed with IC1-3 powders indicated non-inhibitory properties against *Geotrichum candidum* with the inhibitory areas being less than 9.7 mm, which could explain the lack of antifungal effect of ICs after inclusion in EFs.

### 3.4.2. Antiradicalic Activity of EF (DPPH Radical Scavenging Activity)

The phenolic hydroxyl group in thymol has a very high antioxidant activity by inhibiting free radicals [30]. TEO is rich in many flavonoids and phenolic antioxidants like zeaxanthin, lutein, pigenin, naringenin, luteolin, and thymonin [51]. Unencapsulated TEO has a significantly higher ($p < 0.05$) content of the free phenolic hydroxyl group with bioactive properties in comparison with the encapsulated one where the OH groups interacted with β-CD. The same findings were reported by Cruz-Valenzuela et al. [52] for encapsulated cinnamon leaf oil in β-cyclodextrin.

The EF/TEO had the highest antiradical activity ($89.032 \pm 8.9\%$). There are no significant differences between the films with ICs, the values being slightly lower compared to EF/TEO ($p < 0.05$) in inhibiting the DPPH free radicals. For films with ICs, the values of inhibition are between 64.76 and 69.55% (Figure 7). It can be noticed that EF/β-CD also displayed antiradical activity. Other researchers also observed that β-CD could display radical scavenging activity favoring the inclusion of the free radical inside the β-CD cavity [53].

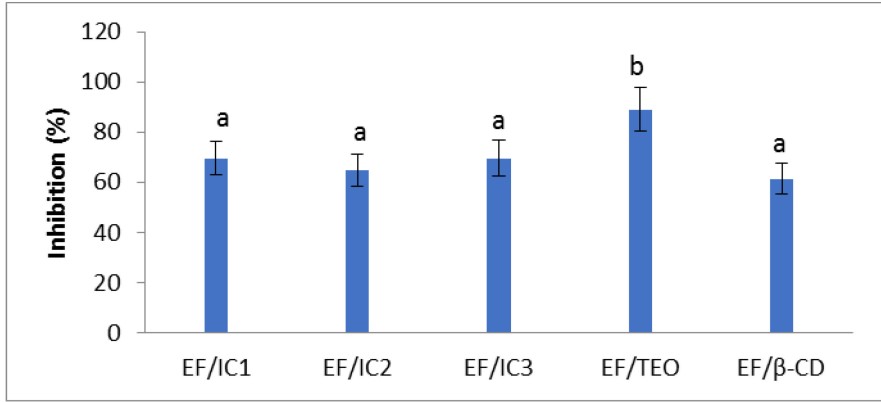

**Figure 7.** Antiradical activity of films. The values obtained are the mean ± SD of three analyses. Different letters obtained by the ANOVA method and Tukey's post hoc test at $p < 0.05$ indicate significant differences between samples.

Although TEO has been encapsulated in β-CD powders that were added into a whey film matrix, the antiradical activity of EF/ICs remained high, suggesting that the films could play an important antioxidant role in food packaging. In their study, Bai et al. [30] also observed that chitosan film with 2% inclusion complex with thymol possessed scavenging capacity of $77.8 \pm 0.7\%$, against DPPH free radicals and the antiradical properties of the films, was concentration-dependent. In their study, Zeid et al. [54] found antiradical

activity values of 79.4%, 84.3%, and 83.3% for films containing thyme, rosemary, and oregano essential oils, respectively. These results have the same order of magnitude as ours, considering the different plants, climate, soil used as well as films' composition.

### 3.4.3. Release Study

Two food simulants (95% ethanol and water) with different polarities were used to study the release of VOCs encapsulated in β-CD from EFs. Diffusion takes place by, first, the migration of outer solution into the polymeric matrix, that produces a weakening of the network, followed by the VOCs' diffusion into the outer solution. The process takes place until thermodynamic equilibrium is achieved [33].

As can be seen in Figure 8a, diffusion is more intense in the ethanolic solution than in water (Figure 8b), where the process takes place at a slower rate. The release rate after 48 h, at 25 °C, in water, is reaching almost 40% in comparison with nearly 70% in ethanol. In comparison with unencapsulated EF/TEO, the release rate, after 48 h, in ethanol from EF/IC with TEO encapsulated in β-CD was reduced by 1.85-, 3.73-, and 4.71-fold in EF/IC1, EF/IC2, and EF/IC3, respectively (Figure 8a). The same findings were reported by Kfoury et al. [33]. Figure 8a shows a lower release rate from EF/IC2 and EF/IC3 compared to EF/IC1 and this could be explained by the reduced concentration of the main volatiles contained in the IC powders, associated with encapsulation efficiency (Table 1). The time to reach a quasi-equilibrium in ethanol simulant from the EF/ICs kept at 25 °C is approximately 24 h and, after 48 h, the films started to be slowly deteriorated by the simulant solution.

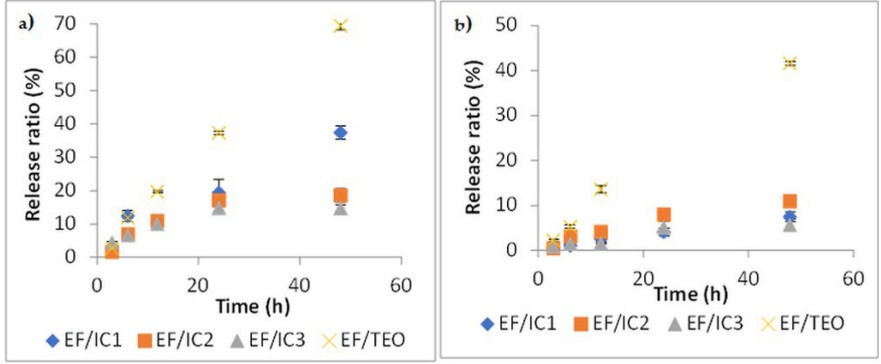

**Figure 8.** Release kinetics of VOCs from EF/ICs at 25 °C in two simulants: (**a**) into 95% ethanol and (**b**) into distilled water. Values represented in the graphs are the mean of three replicates ± SD.

A possible explanation for the higher release rates from EF/ICs in ethanolic solution compared to water is related to the higher solubility in ethanol of thymol, carvacrol, α-pinene, and 3-carene, which are the terpenes with the highest concentration in the IC powders (Table 1). In water, the equilibrium of EF/ICs was reached after 48 h at 25 °C with a maximum release rate of 10% in EF/IC2. In both simulants, EF/TEO showed a significantly higher release rate in comparison with EF/ICs, demonstrating the protective effect of films in which TEO was encapsulated in β-CD.

This indicated that encapsulation within β-CD considerably reduced the volatility of terpenes and allowed their gradual release in both simulants; however, the polarity of the solvent influenced the release rates.

### 4. Conclusions

Encapsulation of TEO in ICs favored the presence of thymol as the main volatile compound in powders. The analysis of films' microstructure showed ICs with diameters in the nanometer range. Films with TEO encapsulated in β-CD displayed lower WVP than films with unencapsulated TEO. The transparency of the films with ICs did not decrease and, in specific cases, was improved compared to EF/TEO film. The FTIR spectra showed

that EF/ICs have different vibration peaks compared to EF with β-CD. The reported results indicate that edible films with ICs could protect the main active volatiles against bacteria and yeasts, but also display good antioxidant activity. Nevertheless, there was a significant reduction in antimould activity in the films with TEO/β-CD compared to EF/TEO. No significant differences were noticed among the three studied formulae of films regarding swelling index, WVP, and antimicrobial and antioxidant activities, probably due to interactions between the whey-based biopolymer and TEO/β-CD inclusion complexes. Encapsulation allowed a better retention of the volatiles in the film structure and a gradual release of the volatiles, with slower release rates in water than in 95% ethanol solution.

These films could be used as packaging for compatible food matrices, i.e., in the cheese industry, where they could exert a protective antioxidant and antimicrobial activity together with a flavoring role.

**Author Contributions:** Conceptualization, D.B., I.B. and C.D.; methodology, A.L.D., A.C. and A.B.; software, A.L.D., A.C. and R.D.; validation, I.B. and D.B.; formal analysis, D.B., R.D. and I.B.; investigation, I.B.; writing—original draft preparation, A.L.D. and I.B.; writing—review and editing, A.L.D., I.B. and D.B; visualization, A.C.; supervision, D.B. All authors have read and agreed to the published version of the manuscript.

**Funding:** This research received no external funding.

**Institutional Review Board Statement:** Not applicable.

**Informed Consent Statement:** Not applicable.

**Data Availability Statement:** Not applicable.

**Conflicts of Interest:** The authors declare no conflict of interest.

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
