# Peer review of "Controlled Release of β-CD-Encapsulated Thyme Essential Oil from Whey Protein Edible Packaging"

_coatings, doi:10.3390/coatings13030508_

Round 1

Reviewer 1 Report

Manuscript 2222188

Journal Coatings

Title Controlled Release of β-CD -encapsulated Thyme Essential Oil from Whey Protein Edible Packaging

The manuscript entitled “Controlled Release of β-CD -encapsulated Thyme Essential Oil from Whey Protein Edible Packaging” describes the production of whey protein films including inclusion complexes with thyme essential oil, the characterization of the physico-chemical properties of the films, and the antioxidant and antimicrobial activity of the edible packaging. However, authors did not find an improvement of antimicrobial and antioxidant activity with the encapsulated thyme essential oil. Authors did not analyze the release of volatiles from edible packaging. A major revision is suggested. Please follow the comments in the file.

Author Response

Reviewer 1

We wish to thank the reviewer for for the constructive observations that enabled us improve the manuscript.

Manuscript 2222188

Journal Coatings

Title Controlled Release of β-CD -encapsulated Thyme Essential Oil from Whey Protein Edible Packaging

The manuscript entitled “Controlled Release of β-CD -encapsulated Thyme Essential Oil from Whey Protein Edible Packaging” describes the production of whey protein films including inclusion complexes with thyme essential oil, the characterization of the physico-chemical properties of the films, and the antioxidant and antimicrobial activity of the edible packaging. However, authors did not find an improvement of antimicrobial and antioxidant activity with the encapsulated thyme essential oil. Authors did not analyze the release of volatiles from edible packaging. A major revision is suggested. Please follow the comments below.

L2-3 Authors did not analyze the release of volatiles from edible packaging. Authors analyzed volatiles only in inclusion complexes. Please revise the title.

We wish the thank the reviewer for the opportunity given to improve the manuscript. It is true that we have assessed only the volatiles present in the inclusion complexes. However, to respond  the reviewer`s suggestions we have studied the controlled release of volatiles from edible films in kinetics experiments using spectrometry at 370 nm, in two simulants. (Sukhtezari et al., 2017- Carbohydrate Polymers). Lines 599-629

L12-25 Rewrite the abstract. Several sentences are confusing. Moreover, results related to antimicrobial and antioxidant activity should be included.

The abstract was re-written in a much clear form. The antioxidant and antimicrobial activity of the EFs was specified. Lines 13-25

L40-50 Please be more specific. Add results related to the antimicrobial and antioxidant activity of edible packaging including thyme essential oil for example.

The following paragraph was added in lines 62-69, according to suggestions made by the reviewer:

For example, Liu et al., 2021 reported that konjac glucomannan EF with 1.6% TEO exerted antibacterial effect, higher against Gram positive bacteria (L. monocytogenes and S. aureus) than Gram negative bacteria (E. coli O157:H7) and a 1.5 increase of the antioxidant effect of the EF with 1.6% TEO compared to the control film, without TEO, due to the high phenolic content present in TEO. Bleoanca et al, 2020 tested the antimicrobial activity by vapor phase test of 2.5% (w/w) TEO volatiles in whey protein EFs and concluded their effective antimicrobial activity against Bacillus subtilis, Geotrichum candidum and Torulopsis stellata.

L50-52 Please introduce the techniques used to protect essential oils (and in particular thyme essential oil and its compounds) and useful to produce active packaging. The production of composite materials, the formation of nanoemulsions, and the encapsulation should be mentioned. The papers

doi.org/10.1016/j.heliyon.2022.e09551, doi.org/10.1016/j.ijbiomac.2020.02.066,

doi.org/10.1016/j.foodhyd.2020.106570, doi.org/10.1016/j.fbio.2021.101230 are suggested for your analysis and discussion.

The following paragraph was added in lines 71-77, as suggested by the reviewer:

Several techniques are available for EO protection, already used for active packaging, like their inclusion in composite materials (e.g. zeolite-thymol antimicrobial composites (Cometa et al, 2022), nanoemulsion formation (e.g.  EF/TEO nanoemulsion used for zucchini active coating- Bleoanca et al. 2022, thymol nanoemulsion included in gelatin films- Li et al., 2020) or encapsulation (e.g. edible coating with encapsulated TEO in liposomal chitosan applied to cheese- Al-Moghazy, 2021), antimicrobial packaging with coriander EO encapsulated in cyclodextrin nano-sponges-Silva, 2019).

L52-56 Please better describe the encapsulation strategies of essential oils (e.g., spray-drying, formation of inclusion complexes, inclusion in liposomes and so on) with relevant references. The paper doi.org/10.1016/j.foodcont.2022.108883 is suggested for your analysis.  

The following paragraph was added at lines 80-82, according to reviewer suggestion:

Several encapsulation techniques have been developed and tested on EOs, like spray-drying (Jafari et al., 2021), complex coacervation (Napiórkowska A, Kurek M., 2022), electrospinning (Linlin et al., 2020), liposome formation, solid lipid nanoparticles produced by high pressure/ high-speed homogenization (Mukurumbira et al., 2022).

β-cyclodextrin (β-CD) has the capacity of selectively encapsulating lipophilic molecules of EOs inside the truncated cone surface, thus protecting their bioactive properties, being one of the most used encapsulant material.

L64-65 Please expand this part with relevant references  

The following text was inserted at lines 93-100 as suggested by reviewer:

Whey EFs can be functionalized with EOs and blended with other biopolymers to form composite materials with improved properties. For example, Kontogianni, 2022 functionalized whey protein concentrate EF with aqueous rosemary and sage extract and successfully applied it to soft cheese stored 60 days of cold storage at 4°C. An updated review on active ingredients used on whey protein EFs is presented by the research group of Kandasami, 2021. Another reported successful example refers to the antioxidant nanocomposite packaging obtained by thermoplastic extrusion of corn starch and whey protein, functionalized with rosemary EO (Azevedo, 2019).

L70-73 Rewrite the objective of the work. All the experimental steps should be introduced, including the antimicrobial and antioxidant activity of the films.  

The objectives of the work were reformulated and all the experiments are now included in lines 105-111.

L91-92 How the mass ratio was selected? Please explain

This part of the research study that explains why this mass ratio was used, is currently under review, sent to another journal and for the moment since it`s not yet accepted, we cannot refer to/cite our work. However, we have added the following statement in lines 131-132:

We have been optimized previously the TEO/β-CD mass ratio within the ICs powders and estimated the efficacy and yield of the encapsulation process (data not shown).

L88-99 Why authors did not determine the yield and the encapsulation efficiency? Please explain  

The same explanation is included in lines 131-132

L115-120 Why 2.5% w/w was used? Please explain.  

Based on the previous published research results, our group has established that 2.5 % is a reasonably good amount for inclusion in films` structure (Bleoanca, Enachi, Borda, 2020).

L143-144 Please add the part related to the identification and quantification of different compounds.  

The suggested information was added in lines 184-186 :

“The volatile organic compounds (VOCs) were tentatively identified using NIST 08 library database available within Xcalibur software and reported as percentage of the individual peak area from the total VOCs area.”  

L181-183 Please check the references  

The authors thank the reviewer for the observation.  We have carefully revised reference section throughout the manuscript.

L211-212 Why authors did not determine the release of thyme oil compounds from edible packaging (e.g., using food simulants)? Please explain  

Thank you for your suggestion. We have reconsidered this part and included the controlled release in two simulants. A new part of research was added in lines 599-629 referring at controlled release studies.

L212-223 Please better describe the method used (inoculum preparation, incubation conditions, and all the experimental steps)  

2.6.1. Antimicrobial activity of EFs, lines 262-267:

EFs containing encapsulated TEO (IC1, IC2 and IC3) were analyzed for antimicrobial activity by disk diffusion assay [16] against three test microorganisms, Bacillus cereus ATCC 10876, Geotrichum candidum and Rhodotorula glutinis from MIUG collection- Dunarea de Jos University of Galati, Romania. Plate count agar was used for antibacterial assay and rose- bengal chloramphenicol agar (Oxoid Ltd., UK) was used for anti- yeast and fungi growth evaluation. On the culture media previously inoculated with test microorganisms (approx. 106 CFU/mL) films with 5 cm diameter were deposited. Following incubation at either 37°C for bacteria or 27°C for yeast and fungi, inhibition diameters were measured with a digital caliper (Burg Wachter, Germany). The results of three independent experiments are reported as average values of the extent of inhibition zones ± standard deviation.

L224-239 Why only DPPH test was used? Generally, two or more antioxidant tests should be used (e.g., ABTS, FRAP, ORAC and so on). Please explain

It is true that more than one test is usually applied to better quantify the antiradical activity, because each test has its own advantages and disadvantages reviewed by literature, however in the current study the authors consider that a relative comparison between the EFs is satisfying and it is not mandatory to quantify the absolute antiradical activity by different methods.

L240-250 Please merge in one section

The statistical methods are now merged into one single section, currently lines 303-313.

L252-271 No results are presented in this section. Please move this part in a different point of the manuscript or delete.

A part of the comments were deleted (lines 315-333) and a small part was moved under 3.1 (lines 336-347). The part referring at “Volatile Fingerprint of TEO/β-CD inclusion complexes”

L264-265 Please check these values. Authors should use those reported at lines 91-92. DB

The values were reported in a similar manner in line 340 with the ones reported line 130.

L284-287 Please discuss these results  

Details were added in lines 351-356.

The high content of thymol in IC1 could be related with a higher encapsulation rate in IC1 compared with IC2 and IC3. Other major compounds were o-cymene, carvacrol 3- carene and α-pinene. The different % of volatiles in each IC could suggest different properties of the powders depending on the concentration and bioactivity of the active molecules

L299-300 Revise the references. The MDPI style should be used  

The authors thank the reviewer for identifying the use of a different style compared to the one requested and we have corrected.

L315-317 Are these peaks different among different samples? Please explain in the text  

Explanation was added in the legend for the Loadings graph (Figure 2), line 409.

L375 Replace charged with loaded  

The authors thank the reviewer for suggestion, we replaced the word, line 464.

L386-387 Here and throughout the manuscript add the footnotes in the figure caption  

The footnotes for the figure captions was modified (i.e. Figure 3, lines 473-476).

L416 Please check the reference. Please use the MDPI style.  

The authors thank the reviewer for identifying the use of a different style compared to the one requested and we have corrected.

L437-439 Rewrite. It is not correct in English

The following paragraph was reformulated as following and inserted in lines 536-538:

All EFs with unencapsulated TEO had antimicrobial effect against all test microorganisms, the inhibition diameters ranging from 20.71 mm (R. glutinis) to 47.57 mm (G. candidum), signifficantly different from one to another (Table 4).

Table 4 Why encapsulated TEO showed lower antimicrobial effect than unencapsulated TEO? Please discuss this result in the text.  

The following text was added to the original manuscript, in lines 545-549, according to reviewer’s suggestion:

Antimicrobial activity of encapsulated materials is usually higher than that of free form (Matouskova, 2016), but there are also exceptions (Bilenler et al., 2015) reporting contradictory results, which is also this research case. The reduced antimicrobial activity exerted by EF/ICs could be explained by the longer time needed for releasing TEO from ICs particles to exert antimicrobial activity (Bilenler et al., 2015). It could be also explained by the binding of βCD with whey protein matrix in EF (Kfoury, M., 2015)

Figure 7 Why encapsulated TEO showed lower antioxidant effect than unencapsulated TEO? Please discuss this result in the text.  

A new paragraph was added in lines 567-571:

Unecapsulated TEO have a significantly higher (p < 0.05) content of free phenolic hydroxyl group with bioactive properties in comparison with encapsulated one where OH groups interacted with β-CD. The same findings were reported by Cruz-Valenzuela et al. (2019) for encapsulated cinnamon leaf oil in β-cyclodextrin

L475-479 Please check the sentences. Repeated sentences are reported

The authors thank reviewer for identifying the repetition, which was unintentional, we have deleted one phrase.

L480-489 Rewrite the conclusion section. The main findings should be reported. Moreover, the potential applications of your results should be highlighted. Finally, a perspective could be added to this section.

The conclusion part was re-written and perspective was added in lines 632-650.

References (L499-L582) Please check the references. Few references are repeated (e.g., 1 and 11). Please add the suggested papers and revise the numbering in the text.

References were carefully reviewed throughout the manuscript and the duplicated reference was deleted.

Reviewer 2 Report

The article entitled “Controlled Release of β-CD -encapsulated Thyme Essential Oil from Whey Protein Edible Packaging”. The authors developed whey protein-based edible films loaded with essential oil. The article is interesting and the experiments are accurately described. I suggest this article can be accepted after some modifications.

1.      Line 296: It should be explained why presence of TEO droplets turns lighter spheres into the darker.

2.      It is very difficult to identify each sample in Fig.2a.

3.      Line 313: Reference should be provided.

4.      Table 2: The thickness should be placed before DM.

5.      Line 335: Why is DM important to properties of films?

6.      Table 3: Why was EF/β-CD not measured?

7.      Figure 5: Why did EF/TEO exhibit higher water vapor permeability than EF/β-CD?

8.      Line 447: Why did all films with unencapsulated TEO show stronger inhibition than encapsulated one?

9.      Figure 7: Why could EF/β-CD exhibit antiradical activity?

Author Response

Reviewer 2

We wish to thank the reviewer for the constructive observations that enabled us improve the manuscript

The article entitled “Controlled Release of β-CD -encapsulated Thyme Essential Oil from Whey Protein Edible Packaging”. The authors developed whey protein-based edible films loaded with essential oil. The article is interesting and the experiments are accurately described. I suggest this article can be accepted after some modifications.

  1. Line 296: It should be explained why presence of TEO droplets turns lighter spheres into the darker.

The text in the manuscript at line 295-296 is: “Microstructure evaluation by scanning the cross-sections of EF/TEO (Figure 1a) indicates the presence of TEO droplets as lighter spheres into the darker EF matrix”.

It seems to be a misunderstanding. However, in order to clarify the situation we underline that there is no change in TEO droplets from lighter into darker shade. The TEO droplets appear in a lighter shade, while the EF matrix appears as a darker shade than the TEO droplets.  Also, the following text was added in the manuscript at lines 376-380:

During the electron beam interaction and due to their different chemical natures, the TEO, ICs and EF appear in various gray tones and morphologies in the SEM images. Therefore, three types of structure can be observed: the TEO components as whiter spheres (Fig. 1a) and the IC inclusions like rods (Fig. 1 c) – both having the crystalline structure – which are incorporated/encapsulated into the EF matrix, as a continuous, homogeneous film (amorphous structure).

  1. It is very difficult to identify each sample in Fig.2a.

The resolution of the figure has been improved and now a larger Figure 2a was inserted.

  1. Line 313: Reference should be provided.  

A new reference was added in lines 750 and 398.

Badea, M.L.; Iconaru, S.L.; Groza, A.; Chifiriuc, M.C.; Beuran, M.; Predoi, D. Peppermint Essential Oil-Doped Hydroxyapatite Nanoparticles with Antimicrobial Properties. Molecules 201924, 2169. https://doi.org/10.3390/molecules24112169

  1. Table 2: The thickness should be placed before DM.

The authors thank the reviewer for suggestion. We changed the order of analysed parameter in the table 2, lines 416-417.

  1. Line 335: Why is DM important to properties of films?

We added the following explanation in manuscript in lines 420-423:

The dry substance plays an important role in evaluating the performance of the film, since a low value of the dry substance and implicitly a high value of the film's humidity can change the properties of the film and of the food covered by film, due to migration or moisture transferring. (S. Othman, 2017)

  1. Table 3: Why was EF/β-CD not measured?

EF/β-CD was measured and we have completed the table 3 with the suggested results, last row.

  1. Figure 5: Why did EF/TEO exhibit higher water vapor permeability than EF/β-CD?

The authors have explained this behavior in lines 501-505 and reference was also added to justify this explanation “This suprising behaviour was also observed in other studies and its occurrence was explained by the increased density in the constituent elements or their concentration in the film that resulted in changes of the WVP [33]. Thus, formation of a dense network films due to the introduction of β-CD powders could have reduced the volume occupied by water molecules.”

  1. Line 447: Why did all films with unencapsulated TEO show stronger inhibition than encapsulated one?

The explanation resides most probably in the diffusion rate of TEO in the growth medium: unencapsulated TEO rapidly diffuses into the growth medium exerting stronger inhibition compared to the encapsulated ones.

The following text was added to the original manuscript, in lines 545-549, to reviewer’s suggestion:

Antimicrobial activity of encapsulated materials is usually higher than that of free form (Matouskova, 2016), but there are also exceptions (Bilenler et al., 2015), which is also this research case, reporting vice versa results. The reduced antimicrobial activity exerted by EF/ICs could be explained by the longer time needed for releasing TEO from ICs particles to exert antimicrobial activity (Bilenler et al., 2015).

  1. Figure 7: Why could EF/β-CD exhibit antiradical activity?

A possible explanation was added in lines 575-577:

Other researchers also observed that β-CD could display radical scavenging activity favoring the inclusion of the free radical inside the β-CD cavity (Kfoury, 2015).

Round 2

Reviewer 1 Report

Authors revised the original version, addressing large part of the reviewer's comments. Minor comments are reported below:

L25-26 Please add a conclusion in the abstract

L599-629 Please revise the English in this part. Several sentences are confusing and not correct in English.

Figure 8 Please add a statistical analysis in the figure and add the related text in the figure caption.

Author Response

Thank you for carefully revising our manuscript. We have considered the reviewer`s suggestions and we made the following changes:

L25-26 Please add a conclusion in the abstract:

The following sentence was added to the abstract (lines 25-26):

These results have demonstrated the properties of EF/ICs with TEO/β-CD as bioactive packaging systems for foods.

L599-629 Please revise the English in this part. Several sentences are confusing and not correct in English.

Confusing sentences were rephrased as following (lines 605-636):

Two food simulants (95 % ethanol and water) with different polarities were used to study the release of VOCs encapsulated in β-CD from EFs. Diffusion takes place by firstly migration of outer solution into the polymeric matrix, that produces a weaking of the network followed by the VOCs diffusion into the outer solution. The process takes place until thermodynamic equilibrium is achieved [34].

As it can be seen in Figure 8a diffusion is more intense in the ethanolic solution than in water (Figure 8b), where the process takes place slower. The release rate after 48 h, at 25 áµ’C, in water, is reaching almost 40% in comparison with nearly 70% in ethanol. In comparison with unencapsulated EF/TEO, the release rate after 48 h in ethanol from EF/IC with TEO encapsulated in β-CD was reduced, by 1.85, 3.73 and 4.71- fold in EF/IC1, EF/IC2 and EF/IC3, respectively (Figure 8a). The same findings were reported by Kfoury et al. [34]. Figure 8a shows a lower release rate from EF/IC2 and EF/IC3 compared to EF/IC1 and this could be explained by the reduced concentration of the main volatiles contained in the IC powders, associated with encapsulation efficiency (Table 1). The time to reach a quasi-equilibrium in ethanol, from the EF/ICs kept at 25 áµ’C is approximately 24 h, and after 48 h the films started to be slowly deteriorated by the simulant solution.

A possible explanation of the higher release rates from EF/ICs in ethanolic solution compared to water,  is related with the higher solubility in ethanol  of thymol, carvacrol, α-pinene, 3-carene, which are the terpenes with the highest concentration in the IC powders (Table 1). In water the equilibrium of EF/ICs was reached after 48 h at 25 áµ’C with a maximum release rate of 10 % in EF/IC2. In both simulants EF/TEO showed a significantly higher release rate in comparison with EF/ICs, demonstrating the protective effect of films in which TEO was encapsulated in β-CD.

This indicated that encapsulation within β-CD, considerably reduced the volatility of terpenes and allowed their gradually release in both simulants, however the polarity of the solvent influenced the release rates.

Figure 8 Please add a statistical analysis in the figure and add the related text in the figure caption.

In figure 8 the data points are represented with SD error bars; therefore the following information was added in the figure caption (lines 603-604):

Values represented in the graphs are the mean of three replicates ± SD.